# On the Global Convergence of Online RLHF with Neural Parametrization

## Abstract

The importance of Reinforcement Learning from Human Feedback (RLHF) in aligning large language models (LLMs) with human values cannot be overstated. RLHF is a three-stage process that includes supervised fine-tuning (SFT), reward learning, and policy learning. Although there are several offline and online approaches to aligning LLMs, they often suffer from distribution shift issues. These issues arise from the inability to accurately capture the distributional interdependence between the reward learning and policy learning stages. Consequently, this has led to various approximated approaches, but the theoretical insights and motivations remain largely limited to tabular settings, which do not hold in practice. This gap between theoretical insights and practical implementations is critical. It is challenging to address this gap as it requires analyzing the performance of AI alignment algorithms in neural network-parameterized settings. Although bi-level formulations have shown promise in addressing distribution shift issues, they suffer from the hyper-gradient problem, and current approaches lack efficient algorithms to solve this. In this work, we tackle these challenges employing the bi-level formulation laid out in Kwon et al. (2024) along with the assumption *Weak Gradient Domination* to demonstrate convergence in an RLHF setup, obtaining a sample complexity of $\epsilon^{-\frac{7}{2}}$. Our key contributions are twofold: (i) We propose a bi-level formulation for AI alignment in parameterized settings and introduce a first-order approach to solve this problem. (ii) We analyze the theoretical convergence rates of the proposed algorithm and derive state-of-the-art bounds. To the best of our knowledge, this is the first work to establish convergence rate bounds and global optimality for the RLHF framework in neural network-parameterized settings. Our contributions are primarily theoretical, providing crucial insights for future practical implementations.

## 1 Introduction

As state-of-the-art artificial intelligence (AI) systems begin to exceed human performance in numerous tasks, aligning these systems with human values and ethics becomes critically important. Reinforcement Learning from Human Feedback (RLHF) has emerged as an effective strategy for achieving AI alignment. Models such as Google's Gemini, Anthropic's Claude, and OpenAI's GPT-4 have shown safe and aligned behaviors using this approach (Dai et al., 2023). However, most current RLHF-based alignment approaches are offline (Ouyang et al., 2022a; Rafailov et al., 2023). By offline it means that the preference datasets are collected offline to fine-tune the supervised fine-tuned (SFT) model. This method introduces a fundamental bottleneck in improving performance through RLHF because it depends on the data distribution coverage of possible query-response pairs, leading to suboptimal alignment.

The existing literature has proposed online RLHF methods (Yuan et al., 2024; Guo et al., 2024a; Sharma et al., 2024a; Lee et al., 2023) that address the issue of static preference datasets inherent in offline settings. However, online RLHF introduces its own set of fundamental challenges, such as determining the optimal way to generate new responses and collect preference feedback. These challenges are generally addressed in current research (Sharma et al., 2024a; Lee et al., 2023) by utilizing the LLM being trained and employing preference oracles for feedback. Interestingly, this approach leads to a well-known distribution shift due to the interdependence between the reward

learning stage and response generation (Guo et al., 2024b; Chakraborty et al., 2024a; Shen et al., 2024). This issue has been effectively captured by recently proposed bi-level optimization frameworks for alignment (Chakraborty et al., 2024a; Ding et al., 2024a).

Interestingly, all of the previously discussed approaches, to deal with the theoretical aspects of the alignment problem formulation, make a fundamental assumption that the underlying language model is tabular. This assumption renders the alignment problem strongly convex, enabling a closed-form solution. Such solutions are commonly utilized in existing methods to derive the final alignment algorithms, for example, DPO, IPO, SLiC, SPIN, and SAIL. However, in real-world applications, this is not the case; language models are invariably parameterized using neural networks. This neural parameterization removes the strong convexity of the problem, making existing theoretical solutions inapplicable, even though they may work empirically. This discrepancy reveals a significant gap between theory and practice in RLHF alignment approaches.

Existing theoretical analyses of RLHF have several significant limitations. For example, Wang et al. (2023) establishes the optimality of a proposed algorithm within the RLHF framework but does not include a parameterized reward model, which is a standard component in modern RLHF methods (Hu et al., 2024). Similarly, Das et al. (2024) achieves convergence but only for linear reward functions —a condition not met in practical RLHF implementations. In another instance, Du et al. (2024) allows neural networks to represent the value function but still restricts the reward function to be linear. Additionally, the sample complexity they achieve is on the order of $\epsilon^{-18}$, rendering it impractical for real-world applications. Li et al. (2024) prove convergence within the RLHF setup under the assumption that the reward function resides in a latent low-dimensional subspace. However, their convergence results pertain to the parameters of the neural networks representing the reward and value functions, rather than to the number of algorithm iterations, and thus do not provide a sample complexity analysis. Furthermore, Xie et al. (2024) demonstrate the convergence of the DPO algorithm (which also utilizes the tabular policy assumption to derive the DPO loss), and requires access to minimizers of non-convex loss functions at each iteration—assumptions that are difficult to satisfy in practice. Aside from these issues, none of the existing studies have theoretically investigated the online RLHF problem formulated using bilevel optimization.

We tackle this challenge by developing the first global convergence analysis for the online RLHF method. Our work employs neural network parameterizations and provides a sample complexity analysis. We summarize our contributions as follows.

- **First order algorithm for parameterized bi-level formulation of online RLHF problem.** We consider the parameterized form of the bilevel formulation of the online RLHF problem (Chakraborty et al., 2024a; Ding et al., 2024a) and develop a first-order approach to solve the problem. True to our best knowledge, this algorithm is novel and is derived without the tabular policy assumption which is common in the existing literature.
- **Sample complexity analysis under neural parameterized settings.** We derive the first sample complexity bounds for online RLHF in parameterized settings. In order to obtain the sample complexity of the above algorithm, we use the *weak gradient domination assumption* on the reward loss function. Note that this is a weaker assumption than the one used in Chen et al. (2024) (which relies upon the *strong gradient domination assumption*). This allows us to move beyond the linear reward assumption required in existing results. We are thus able to obtain a sample complexity of $\epsilon^{-\frac{3}{2}}$, which is state of the art in online RLHF domain.

## 2 RELATED WORKS

**AI Alignment via RLHF.** The use of human preferences in reinforcement learning (RL) has been explored for over a decade (Jain et al., 2013; Busa-Fekete et al., 2014) and has subsequently been integrated with Deep RL (Christiano et al., 2017). This approach has demonstrated empirical success in areas like robotics (Jain et al., 2013; Abramson et al., 2022; Ding et al., 2023) and fine-tuning large language models (Ziegler et al., 2020; Ouyang et al., 2022b; Bai et al., 2022). Majority of current research in LLM alignment focusses on offline methods such as (Ouyang et al., 2022b; Chakraborty et al., 2024c). However, there has been a recent push to implement online methods (Christiano et al., 2017; Lee et al., 2024; Bai et al., 2022). However, these methods suffer from distribution shift and thus lead to sub-optimal performance (Sharma et al., 2024b). Works such as Chakraborty et al.

Table 1: This table summarizes the features of different RLHF results. Our result is the first to provide sample complexity results of RLHF for an MDP accounts for distribution shift.

| References | Accounts for Distribution Shift | Efficient First Order Implementation | General Function Approximation for Reward Function | Sample Complexity |
|---|---|---|---|---|
| Wang et al. (2023) | ✗ | ✗ | ✗ | $\tilde{\mathcal{O}}(\epsilon^{-2})$ |
| Das et al. (2024) | ✗ | ✗ | ✗ | $\tilde{\mathcal{O}}(\epsilon^{-2})$ |
| Li et al. (2024) | ✗ | ✗ | ✓ | N/A |
| Xie et al. (2024) | ✗ | ✗ | ✓ | $\tilde{\mathcal{O}}(\epsilon^{-2})$ |
| Du et al. (2024) | ✗ | ✓ | ✗ | $\tilde{\mathcal{O}}(\epsilon^{-18})$ |
| This Work | ✓ | ✓ | ✓ | $\tilde{\mathcal{O}}(\epsilon^{-\frac{7}{2}})$ |

(2024c) and Ding et al. (2024b) use a bilevel optimization setup in order to avoid the problem of distribution shift.

**Theoretical Results in RLHF.** Existing literate Novoseller et al. (2020), Pacchiano et al. (2023) and Zhan et al. (2023b) consider tabular or linear MDPs, and assume that the reward is linear in trajectory features. Zhu et al. (2024), Zhan et al. (2023a) consider an offline setting. Further, Wang et al. (2023),Das et al. (2024) are not restricted to tabular space but do require the reward function to be linear.

**Bilevel Optimization.** In general, BO with non-convex lower-level objectives is not computationally tractable without further assumptions, even for the special case of min-max optimization (Daskalakis et al., 2021). Therefore, additional assumptions on the lower-level problem are necessary. The work in Chen et al. (2023) considers several growth conditions for the lower-level objectives (including PL), which guarantee Lipschitz continuity of lower-level solution sets. The work in Shen & Chen (2023b) assumes the PL condition, and studies the complexity of the penalty method in deterministic settings. The work in Arbel & Mairal (2022) studies some asymptotic properties of bi-level optimization. In all these works, underlying assumptions involve some growth conditions of the lower-level problem, which is essential for the continuity of lower-level solution sets.

## 3 PROBLEM FORMULATION

**Markov Decision Process Setup.** We consider a discounted Markov Decision Process (MDP) given by the tuple $\mathcal{M} := (\mathcal{S}, \mathcal{A}, P, R, \gamma)$, where $\mathcal{S}$ is a bounded measurable state space, $\mathcal{A}$ is the set of actions, which is also a bounded measurable space. Note that for our setup, both the state and action space can be infinite. $P : \mathcal{S} \times \mathcal{A} \to \mathcal{P}(\mathcal{S})$ is the probability transition function, $r_\phi : \mathcal{S} \times \mathcal{A} \to [0, 1], (\phi \in \Theta)$ is the parameterised reward function which may be a nonlinear class of functions such as neural networks defined on the state action space. $0 < \gamma < 1$ is the discount factor. A policy $\pi : \mathcal{S} \to \mathcal{P}(\mathcal{A})$ maps a state to a probability distribution over the action space. We define $\rho_\nu^\pi(s)$ as the stationary state distribution induced by the policy $\pi$ starting at state distribution $\nu$ and $\zeta_\nu^\pi(s, a)$ is the corresponding stationary state action distribution defined as $\zeta_\nu^\pi(s, a) = \rho_\nu^\pi(s) \cdot \pi(a|s)$. We can define the state action visitation distribution as $d_\nu^\pi(s, a) = (1-\gamma) \sum_{t=0}^\infty \gamma^t Pr^\pi(s_t = s, a_t = a|(s_0, a_0) \sim \nu)$, where $Pr^\pi(s_t = s, a_t = a|(s_0, a_0) \sim \nu)$ denotes the probability that the state action pair at time $t$ is $(s, a)$ when following the policy $\pi$ with the starting state action distribution of $\nu$.

**Language Model.** In the context of a Large Language Model (LLM), we have a vocabulary set denoted by $\mathcal{V}$, and represent the language model by a policy $\pi$, which takes a sequence of tokens (prompt) as input denoted by $x := \{x_1, x_2, \cdots, x_N\}$, set of prompts denoted by $\mathcal{P}$, and generates the response $y = \{y_1, y_2, \cdots, y_T\}$ in token-by-token fashion. To determine the next token at the $t^{th}$ time-point $y_t$, the input prompt $\mathbf{x}$ and the generated tokens $\mathbf{y}_{<t}$ are fed as input to the language model as a new prompt $[x, y_{<t}]$. The next token is sampled as $y_t \sim \pi(\cdot|[x, y_{<t}])$.

In the RL setup, this can be viewed in the following manner: at every step of the sequence, the prompt plus the generated tokens so far represents the state and the next token represents the action. In the case of LLM's both the state and action space is represented by the vocabulary set $\mathcal{V}$.

In the standard online RLHF setup, the reward function is learned via the maximization of the following two functions.

**Step 1: Reward learning** phase deals with learning the reward function by collecting preferences from some expert feedback or oracle function on the responses generated by the LLM policy optimized from the previous iteration. This is typically done under the Bradley-Terry preference model assumption and is obtained by solving

$$\underset{\phi}{\arg\min} \, \mathbb{E}_{(\tau_1, \tau_2) \sim \mathcal{D}_r} \big[ \kappa((R(\tau_1) - R(\tau_2))) \big], \tag{1}$$

where $\mathcal{D}_r$ represents the dataset of responses $\tau_1, \tau_2$ are the trajectories belonging to dataset $\mathcal{D}_r$ such that $\tau_1$ is preferred over $\tau_2$ and $R_\phi(\tau) = \sum_{i=1}^{H} r_\phi(s_i, a_i)$ denotes the total reward of the trajectory $\tau$ under the reward function $r_\phi$ and $\kappa$ is a loss function.

**Step 2 : Policy optimization/fine tuning** where we learn the LLM policy $\pi_\phi^*$ for a given reward $r_{\phi^*}$ where $\phi^*$ is the minimizer of the loss function given in Equation (1). This is done by solving KL regularized policy optimization problem given as

$$\max_{\pi} \mathbb{E}_{x \sim \mathcal{P}, y \sim \pi(\cdot \mid x)} \big[ r_{\phi^*}(x, y) - \beta \mathbb{D}_{\mathrm{KL}} \big[ \pi(\cdot|x) || \pi_{\mathrm{SFT}}(\cdot|x) \big] \big], \tag{2}$$

where $\beta > 0$ controls the deviation from the base reference policy $\pi_{\mathrm{SFT}}$ and $\mathcal{P}$ is some starting state distribution. This process is repeated over multiple iterations as detailed in (Christiano et al., 2017) by alternatively updating the policy and reward models till convergence.

Although such a setup makes it easier to analyze theoretically, a key drawback of this method as is pointed out in Ding et al. (2024b) is that the data generated for solving Equation (1) is generated from the policy we obtain by optimizing Equation (2). All current state of the art online RLHF methods ignore this interdependence and thus suffer from what is known as *distribution drift*.

## 4    PROPOSED SOLUTION TO DISTRIBUTION DRIFT ISSUE

We first propose a solution to the distribution drift issue. We would like to define the following terms. The action value function for a given policy $\pi$ is given by

$$Q_\phi^\pi(s, a) = \mathbb{E}\left[ \sum_{t=0}^{\infty} \gamma^t r_\phi(s_t, a_t) | s_0 = s, a_0 = a \right], \tag{3}$$

where $a_t \sim \pi(\cdot|s_t)$ and $s_{t+1} \sim P(\cdot|s_t, a_t)$ for $t = \{0, \cdots, \infty\}$. For a discounted MDP, we define the optimal action value functions as

$$Q_\phi^*(s, a) = \sup_{\pi} Q_\phi^\pi(s, a), \quad \forall (s, a) \in \mathcal{S} \times \mathcal{A}. \tag{4}$$

We additionally define the Bellman operator for a policy $\pi$ on a function $Q : \mathcal{S} \times \mathcal{A} \to \mathbb{R}$ is defined as

$$(T^\pi Q)(s, a) = r(s, a) + \gamma \int Q(s', \pi(s')) P(ds'|s, a) \tag{5}$$

Finally, we the expected average return given by

$$J(\lambda, \phi) = \mathbb{E}_{s \sim \nu, a \sim \pi_\lambda(.|s)} [Q_\phi^{\pi_\lambda}(s, a)] \tag{6}$$

where the policy is parameterized as $\{\pi_\lambda, \lambda \in \Lambda\}$ and $\Lambda \subset \mathbb{R}^d$ where $d$ is a positive integer.

Now we describe the bi-level formulation of the RLHF problem. We first define the objective function to evaluate the reward in order to make our proposed algorithm more concrete as follows

$$G(\lambda, \phi) = -\mathbb{E}_{y, \tau_0, \tau_1 \sim \rho_H(\lambda)} (y \cdot P_\phi(\tau_0 > \tau_1) + (1 - y) \cdot (1 - P_\phi(\tau_0 > \tau_1))) \tag{7}$$

Where $\rho_H(\lambda)$ is the distribution of a trajectory of length $H$ by following the policy $\lambda$ and $y$ is the preference which is 1 if trajectory 1 is preferred and 0 if Trajectory 0 is preferred which is drawn from some unknown distribution $\rho$. Also, $P_\phi(\tau_0 > \tau_1)$ is defined as

$$P_\phi(\tau_0 > \tau_1) = \frac{\exp \sum_{h=0}^{H-1} r_\phi(s_h^0, a_h^0)}{\exp \sum_{h=0}^{H-1} r_\phi(s_h^0, a_h^0) + \exp \sum_{h=0}^{H-1} r_\phi(s_h^1, a_h^1)}, \tag{8}$$

In order to account for the distribution drift, Chakraborty et al. (2024c); Ding et al. (2024a) proposed that the RLHF problem is inherently bilevel in nature given by

$$\max_{\phi} \Phi(\phi) = G(\phi, \lambda^*(\phi)) \textbf{ where } \lambda^* \in argmax_{\lambda} J(\lambda, \phi). \tag{9}$$

It has been empirically demonstrated in Chakraborty et al. (2024b) that the bi-level framework for RLHF leads to better performance compared to prior models where the interdependence between the data generating policy and the reward function was ignored. However, there remains an issue of the *hyper-gradient estimation*. This is because if we take the gradient of $G(\phi, \lambda^*(\phi))$, this would involve calculating the gradient of $\lambda^*(\phi)$ with respect to $\phi$ which does not have any known efficient methods of calculation yet. From Kwon et al. (2024), we can reformulate the above optimization as maximizing the proxy objective defined by

$$\Phi_{\sigma}(\phi) = \max_{\lambda} \left( G(\phi, \lambda) + \frac{J(\lambda^*) - J(\lambda, \phi)}{\sigma} \right), \tag{10}$$

where $J(\lambda^*) = \max_{\lambda \in \Lambda} J(\lambda, \phi)$. The key advantage of this formulation is that we do not have to calculate the gradient of $\lambda^*$ with respect to $\phi$. This fact is demonstrated in Shen & Chen (2023a) which proves that the gradient of the proxy objective is given by

$$\nabla \Phi_{\sigma}(\phi) = \nabla G(\phi, \lambda^*(\phi)) + \frac{\nabla J(\lambda^*(\phi), \phi) - \nabla J(\lambda_{\sigma}^*(\phi), \phi)}{\sigma}. \tag{11}$$

Here $\lambda^*(\phi) = argmax_{\lambda}(J(\lambda, \phi)$ and $\lambda_{\sigma}^*(\phi) = argmax_{\lambda}(J(\lambda, \phi) + \sigma G(\phi, \lambda))$. In contrast to existing RLHF approaches, where the reward optimization is done independently of the policy used to generate the reward trajectories, we can see that the gradient of the objective used to evaluate the reward function depends on the policy used to generate the trajectories.

## 5 PROPOSED ALGORITHM

The main Algorithm 1 proceeds in two stages. The first stage is to calculate the gradient given in Equation (11). This requires us to get the estimates of $\lambda^*(\phi)$ and $\lambda_{\sigma}^*(\phi)$ as defined below Equation (11). This is done in the inner for loop of Algorithm 1. We then evaluate the gradient of $\phi_{\sigma}$ as given in Equation (11) and perform the SGD update on Line 13.

We now go over how to estimate the various required gradient terms estimating the gradient in Equation (11). Let us start with $\nabla_{\lambda} J(\lambda, \phi)$, i.e., the derivative of the average expected return $J(\lambda, \theta)$ with respect to the policy parameter. This can be obtained from the policy gradient theorem given by

$$\nabla_{\lambda} J(\lambda, \phi) = \mathbb{E}_{(s,a) \sim d_{\nu}^{\pi_{\lambda}}} [\nabla_{\lambda} \log \pi_{\lambda} Q_{\phi}^{\lambda}], \tag{12}$$

In order to perform the gradient descent at the $k^{th}$ iteration of the inner for-loop and $t^{th}$ iteration of the main for loop of Algorithm 1, the estimate of $Q_{\phi_t}^{\pi_{\lambda_k}}$ is obtained in Algorithm 2 which solves the following optimization

$$\underset{\theta \in \Theta'}{\arg \min} \mathbb{E}_{(s,a)} \left[ (Q_{\phi_t}^{\pi_{\lambda_k}} - Q_{\theta})^2 \right], \tag{13}$$

$\Theta'$ is the space of parameters for the critic neural networks with $D$ layers and $m$ neurons per layer and $Q_{\theta}$ is the neural network corresponding to the parameter $\theta$. This step is equivalent to the critic step in a vanilla actor critic algorithm. Algorithm 2 carries out the DQN procedure which sequentially applies the empirical estimate of the Bellman optimally operator on the current estimate of $Q_{\phi_t}^{\pi_{\lambda_k}}$. We incorporate experience replay and target network, as these are crucial to the convergence of DQN. The target network is updated every $L$ steps, and the replay of the experience is carried out by resampling from the tuples stored in Algorithm 1. Thus, we obtain a sample estimate of the gradient as given in Equation (12).

Now we come to the gradient of $J(\lambda, \phi)$ with respect to the upper-level variable $\phi$. In Lemma 5 it is shown that this can be written as

$$\nabla_{\phi} J(\lambda, \phi) \quad = \quad \sum_{i=1}^{\infty} \gamma^{i-1} \mathbb{E}[\nabla_{\phi} r_{\phi}(s_i, a_i)], \tag{14}$$

---

**Algorithm 1** A first-order approach to parameterized RLHF

---

1: **Input:** $\mathcal{S}, \mathcal{A}, \gamma$, Time Horizon $K \in \mathcal{Z}$, Number of gradient estimation updates $K \in \mathcal{Z}$, sample batch size $n \in \mathcal{Z}$, target network update frequency $L \in \mathcal{Z}$, ,starting state sampling distribution $\nu$, Critic step size $\beta'$, starting policy parameter $\lambda_0$,
2: **for** $t \in \{1, \cdots, T\}$ **do**
3:    **for** $k \in \{0, \cdots, K-1\}$ **do**
4:        Sample $B$ pair of trajectories of length $H$ and the corresponding preferences by following the policy $\pi^{\lambda_k}$ from a starting state distribution $\nu$. Store the trajectories and corresponding preferences.
5:        Sample $n$ tuples $(s, a, r, s')$ by following the policy $\pi^{\lambda_k}$ from a starting state distribution $\nu$ Store the tuples.
6:        Estimate $Q_{\phi_t}^{\lambda_k}$ from Algorithm 2 denoted as $Q^{k,J}$
7:        $d_k = \frac{1}{n} \sum_{i=1}^{n} \nabla \log(\pi(a_i|s_i)) Q_{k,J}(s_i, a_i)$
8:        $d_k' = \frac{\sigma}{n} \sum_{i=1}^{n} \nabla \log(\pi(a_i|s_i)) Q_{k,J}(s_i, a_i) + \frac{1}{B} \sum_{i=1}^{B} \nabla_\lambda \hat{G}_{\tau_i}(\lambda_t^K, \phi_t)$
9:        $\lambda_t^{k+1} = \lambda_t^k - \tau_k \cdot \frac{d_k}{||d_k||}$
10:        $\lambda_t'^{k+1} = \lambda_t'^k - \tau_k' \cdot \frac{d_k'}{||d_k'||}$
11:    **end for**
12:    Sample $B$ pair of trajectories of length $H$ and the corresponding preferences by following the policy $\pi^{\lambda_k}$ from a starting state distribution $\nu$. Store the trajectories and corresponding preferences.
13:    $d_t = \frac{1}{B} \sum_{i=1}^{B} \nabla_\phi \hat{G}_{\tau_i}(\lambda_t^K, \phi_t) + \frac{\nabla_\phi \hat{J}(\lambda_t^K, \phi_t) - \nabla_\phi \hat{J}(\lambda_t'^K, \phi_t)}{\sigma}$
14:    $\phi_{t+1} = \phi_t - \eta_t \cdot \frac{d_t}{||d_t||}$
15: **end for**

---

Here, $s_i, a_i$ belong to the distribution of the $i^{th}$ state action pair induced by following the policy $\lambda$. We use a truncated sample estimate of the gradient given in Equation (14) given by

$$\nabla_\phi \hat{J}(\lambda, \phi) = \frac{1}{B} \sum_{i=1}^{B} \sum_{j=1}^{H} \gamma^{i-1} \nabla_\phi r_\phi(s_{i,j}, a_{i,j}), \tag{15}$$

Now, we obtain the gradients of the upper-level objective. We follow the convention laid out in Chakraborty et al. (2024b), where the gradient of upper level objective with respect to the upper level variable $\phi$ will be

$$\nabla_\phi G(\phi, \lambda) = \mathbb{E}\left[\nabla_\phi U(\phi)\right], \tag{16}$$

Further, the gradient of the upper-level objective with respect to the lower-level variable $\lambda$ is

$$\nabla_\lambda G(\phi, \lambda) = \mathbb{E}\left[U(\phi) \cdot \sum_{h=0}^{H-1} \nabla_\lambda \log \pi_\lambda(a_h|s_h)\right]. \tag{17}$$

where $U(\phi) = y \cdot P_\phi(\tau_0 > \tau_1) + (1-y) \cdot (1 - P_\phi(\tau_0 > \tau_1))$. We use the stochastic version of all the gradients in the proposed Equations (16) and (17).

## 6 THEORETICAL ANALYSIS: GLOBAL CONVERGENCE

Now that we have demonstrated an efficient first order RLHF parameterised with general reward parametrization, we want to prove that it is able to estimate the optimal reward function and provide a sample complexity for the same. The square of the optimality gap for the upper level loss function $\Phi$ is chosen as it is the loss function minimized for existing RLHF methods (Wang et al., 2023), (Das et al., 2024) as well as bi-level optimization works (Chen et al., 2023).

Before we present the main theorem, we want to state the assumptions we make to obtain the final result.

---

**Algorithm 2** Estimating Q function

1: **for** $j \in \{1, \cdots, J\}$ **do**
2:     Initialize $Q_{\theta_0}$ where $\theta_0$ using a standard Gaussian.
3:     **for** $i \in \{1, \cdots, L\}$ **do**
4:         Sample a tuple $(s_i, a_i, r_i, s_i^{'})$ with equal probability from the stored tuples in Algorithm 1
5:         Sample $a_i^{'}$ using $\pi^{\lambda_k}(.|s_i^{'})$
6:         Set $y_i = r_i + \gamma Q_{k,j-1}(s_i^{'}, a_i^{'})$,
7:         $\theta_i^{'} = \theta_{i-1} + \beta^{'}(y_i - Q_{\theta_i}(s_i, a_i))\nabla Q_{\theta_i}(s_i, a_i)$
8:         $\theta_i = \Gamma_{\theta_0, \frac{1}{(1-\gamma)}}\left(\theta_i^{'}\right)$
9:     **end for**
10: **end for**
11: $Q^{k,j} = Q_{\theta'}$ where $\theta^{'} = \frac{1}{L}\sum_{i=1}^{L}(\theta_i)$
12: Return $Q^{k,j}$

---

## 6.1 ASSUMPTIONS

**Assumption 1.**     *(a) For any $\phi, \phi_1, \phi_2 \in \Theta$, the upper objective $\Phi$ satisfies the weak gradient domination with respect to $\phi$ and smoothness property with respect to $\phi$ as follows*

$$\sqrt{\mu_1}(\Phi^* - \Phi(\phi)) \quad \leq \quad \|\nabla\Phi(\phi)\|, \tag{18}$$
$$\|\nabla\Phi(\phi_1) - \nabla\Phi(\phi_2)\| \quad \leq \quad \alpha_1\|\phi_1 - \phi_2\|, \tag{19}$$

*(b) For any $\phi, \phi_1, \phi_2 \in \Theta$, the function $J_\sigma = J(\lambda, \phi) + \sigma G(\phi, \lambda)$ satisfies the weak gradient domination with respect to $\phi$ and smoothness property with respect to $\phi$ as follows*

$$\sqrt{\mu_2}(J_\sigma^* - J_\sigma(\phi)) \quad \leq \quad \|\nabla J_\sigma(\phi)\|, \tag{20}$$
$$\|\nabla\Phi(\phi_1) - \nabla\Phi(\phi_2)\| \quad \leq \quad \alpha_1\|\phi_1 - \phi_2\|, \tag{21}$$

*where $\Phi^* = \max_{\phi\in\Theta}\Phi(\phi)$ and $\Phi_\sigma^* = \max_{\phi\in\Theta}J_\sigma(\phi)$ and $\mu_1, \mu_2, \alpha_1, \alpha_2 \geq 0$.*

*(c) The maximizer of the upper-level loss function $\Phi$ lies in the function class $\Theta$, i.e., $\phi^* \in \Theta$*

*(d) The variance of the gradient of the upper objective $G(\lambda, \phi)$ is bounded over a joint distribution $\rho$ of the pair of trajectories and corresponding preferences, or*

$$\mathbb{E}\|\nabla_\phi G(\lambda, \phi) - \mathbb{E}\nabla_\phi(G)(\lambda, \phi)\| \leq \sigma_B^2. \tag{22}$$

*Where, $sigma_b$ is a constant.*

These assumptions are similar to standard assumptions required for convergence in bilevel optimizations as in works such as Chen et al. (2024). Our assumptions are weaker since we require the *weak gradient domination condition* which is weaker than the *strong gradient domination condition*. Also, unlike Chen et al. (2024), we do not require the gradient domination or smoothness property on the lower-level objective $J(\lambda, \phi)$. That is implied from Assumptions 2 and 4, which are standard assumptions in the RL literature.

**Assumption 2.** *For any $\lambda, \lambda_1, \lambda_2 \in \Lambda$, $(s, a) \in (\mathcal{S} \times \mathcal{A})$ and $\phi, \phi_1, \phi_2 \in \Theta$, we have*

*(i) $\|\nabla log(\pi_{\lambda_1})(a|s) - \nabla log(\pi_{\lambda_2})(a|s)\| \leq \beta\|\lambda_1 - \lambda_2\|$,*

*(ii) $\|\nabla log(\pi_{\lambda_1})(a|s)\| \leq M_g$,*

*(iii) $\mathbb{E}_{(s,a)\sim d_\nu^{\pi_{\lambda_1}}}(\nabla\log\pi_{\lambda_1}(a|s))(\nabla\log\pi_{\lambda_1}(a|s))^T \succcurlyeq \mu_f I_d$,*

*(iv) $\|\nabla r_{\phi_1}(a|s) - \nabla r_{\phi_2}(a|s)\| \leq \kappa\|\phi_1 - \phi_2\|$,*

*(v) $\|\nabla r_{\phi_1}(a|s)\| \leq M_k$.*

*where $\beta, M_g, \mu_f, \kappa, M_k \geq 0$.*

Such assumptions have been utilized in prior works based on policy gradients such as (Masiha et al., 2022; Fatkhullin et al., 2023) and actor-critic using linear critics such as (Xu et al., 2020), and global convergence results for parameterization of neural critics such as (Fu et al., 2021; Wang et al., 2020), which restrict their analysis to energy-based policies for finite action spaces.

**Assumption 3.** *For any fixed $\phi \in \Theta$ and $\lambda \in \Lambda$, we have that*

$$||\lambda - \lambda_\phi^*| \leq L^{'}.||J(\lambda, \phi) - J(\lambda_\phi^*, \phi)||.$$

*where $\lambda_\phi^* = argmax_{\lambda \in \lambda} J(\lambda.\phi)$ and $L^{'} \geq 0$.*

This assumption is known as the *State Regularity assumption* on the expected return and is used in prior works on nonlinear MDP's such as Tian et al. (2023) and Gaur et al. (2024). This assumption corresponds to the assumption in a linear MDP that the features are linearly independent, but for a nonlinear system. The corresponding assumption for linear MDP's is in many previous works on linear MDP (Wu et al., 2020; Olshevsky & Gharesifard, 2023; Chen & Zhao, 2024; Liu & Olshevsky, 2021).

**Assumption 4.** *For any fixed $\lambda \in \Lambda$ and $\phi \in \Theta$ we have*

$$\mathbb{E}\left(A_\phi^{\pi_\lambda}(s,a) - (1-\gamma)w^*(\lambda)^\top \nabla \log(\pi_\lambda)(a|s)\right)^2. \leq \epsilon_{bias}$$

*Here, the expectation is over $(s,a) \sim d_\nu^{\pi^*}$ where $\pi^*$ is the optimal policy. We also have $w^*(\lambda) = F(\lambda)^\dagger \nabla J(\lambda)$ where $F(\lambda) = \mathbb{E}_{(s,a)\sim d_\nu^{\pi_\lambda}}(\nabla \log \pi_\lambda(a|s))(\nabla \log \pi_\lambda(a|s))^T$ and $A_\phi^{\pi_\lambda}(s,a) = Q_\phi^{\pi_\lambda}(s,a) - \mathbb{E}_{a\sim\pi_\lambda(\cdot|s)}Q_\phi^{\pi_\lambda}(s,a).$*

Wang et al. (2020) proves that this error goes to zero if both the actor and critic are represented by overparametrised neural networks. This assumption allows us to establish the weak gradient bound property for our MDP setup. It is used in policy gradient works such as (Yuan et al., 2022; Masiha et al., 2022; Fatkhullin et al., 2023) to establish last iterate convergence of the estimate of the average return $J(\lambda)$.

**Assumption 5.** *For any fixed $\lambda \in \Lambda$ we have*

$$\min_{\theta_1 \in \Theta'} \mathbb{E}_{s,a\sim\zeta_\nu^{\pi_{\lambda_k}}}\left(Q_{\theta_1}(s,a) - T^{\pi_\lambda}Q_\theta(s,a)\right)^2 \leq \epsilon_{approx}.$$

This assumption ensures that a class of neural networks are able to approximate the function obtained by applying the Bellman operator to a neural network of the same class. Similar assumptions are taken in Fu et al. (2021); Wang et al. (2020); Gaur et al. (2024). In works such as Cayci et al. (2022), stronger assumptions are made wherein the function class used for critic parametrization is assumed to be able to approximate any smooth function.

Before we move on to the main result, we want to state the key lemma proved in Ding et al. (2022), that is used to obtain the last iterate convergence.

**Lemma 1.** *If Assumptions 1 and 5 hold, then for any fixed $\lambda \in \Lambda$ and $\phi \in \Theta$, we have*

$$\sqrt{\mu_3}(J(\lambda^*, \phi) - J(\lambda), \phi) \leq \epsilon^{'} + \|\nabla J(\lambda)\|,$$

*where $\epsilon^{'} = \frac{\mu_f \sqrt{\epsilon_{bias}}}{M_g(1-\gamma)}$ and $\mu_3 = \frac{\mu_f^3}{2M_g^2}$.*

## 6.2 MAIN RESULT

**Theorem 1.** *Suppose Assumptions 2-5 hold and we have $\eta_t = \frac{7}{2t\sqrt{\mu_1}}$, $\tau_k = \frac{7}{2k\sqrt{\mu_3}}$, $\tau_k^{'} = \frac{7}{2k\sqrt{\mu_2}}$ and $\beta^{'} = \frac{1}{\sqrt{L}}$, then from Algorithm 1, we obtain*

$$\begin{aligned}(\Phi^* - \Phi_t)^2 &\leq \tilde{\mathcal{O}}\left(\frac{1}{T^2}\right) + \tilde{\mathcal{O}}\left(\frac{1}{\sigma^2 K^2}\right) + \tilde{\mathcal{O}}\left(\frac{1}{\sigma^2 n}\right) + \tilde{\mathcal{O}}(\epsilon_{bias}) + \tilde{\mathcal{O}}(\epsilon_{approx}^2) \\ &+ \tilde{\mathcal{O}}\left(\frac{1}{B}\right) + \tilde{\mathcal{O}}(\sigma^2) + \tilde{\mathcal{O}}\left(m^{-\frac{1}{6}}D^7\right).\end{aligned} \quad (23)$$

*If we set $\sigma^2 = \tilde{\mathcal{O}}(\epsilon)$, $B = \tilde{\mathcal{O}}(\epsilon)^{-1}$, $n = \tilde{\mathcal{O}}(\epsilon)^{-2}$, $T = \tilde{\mathcal{O}}(\epsilon)^{-\frac{1}{2}}$, $K = \tilde{\mathcal{O}}(\epsilon)^{-1}$. This gives us a sample complexity of $n.K.T + B.K.T = \tilde{\mathcal{O}}(\epsilon)^{-\frac{7}{2}}$.*

Note that the current state-of-the-art sample complexity for a vanilla actor critic with neural parameterization is $\epsilon^{-3}$ achieved in Gaur et al. (2024). Our result has a slightly higher sample complexity of $\epsilon^{-\frac{7}{2}}$. This can be attributed to the fact that in RLHF the reward function is not fixed but has to be estimated from preference data. This makes the problem harder; therefore, it is intuitive that it would have a worst sample complexity.

## 7 PROOF SKETCH OF THEOREM 1

The proof is split into two stages. In the first stage, we show how to obtain the performance gap in the last iteration as a function of the errors incurred in estimating the gradient of $\Phi_\sigma$ in each iteration of Algorithm 1. The second part upper bounds the error incurred in estimating the gradient of the loss function $\Phi_\sigma$.

**Upper Bounding Last Iterate Performance Gap:** Under Assumption 2, from the smoothness of $\Phi$, we have

$$\begin{aligned}-\Phi(\phi_{t+1}) \geq &-\Phi(\phi_t) - \langle \nabla_\phi \Phi(\phi_t), \phi_{t+1} - \phi_t \rangle \\ &+ \alpha_1 ||\phi_{t+1} - \phi_t||^2,\end{aligned} \quad (24)$$

Here, $\alpha_1$ is the smoothness parameter of $\Phi$. Now, using Assumption 3 and the weak gradient domination property of Assumption 1, we obtain the following

$$\begin{aligned}\Phi^* - \Phi(\phi_t) \leq &\left(1 - \frac{\eta_t \sqrt{\mu_1}}{3}\right)(\Phi^* - \Phi(\phi_{t-1})) + ||\nabla_\phi \Phi(\phi_{t-1}) - d_{t-1}|| \\ &+ \alpha_1 ||\phi_t - \phi_{t-1}||^2.\end{aligned} \quad (25)$$

The key step now is recursively applying this condition and and substitute the result back in Equation (25). We thus obtain the following result, the details of which are given in Section A of the Appendix.

$$\Phi(\phi^*) - \Phi(\phi_t) \leq \frac{\eta_1}{t} \sum_{k=0}^{k=t-1} (||\nabla_\phi \Phi(\phi_k) - \nabla_\phi \hat{\Phi}_\sigma(\phi_k)||) + \tilde{\mathcal{O}}\left(\frac{1}{t}\right). \quad (26)$$

**Upper Gradient Estimation Error:** The error in gradient estimation at each iteration $k$ of Algorithm 1 given by $||\nabla_\phi \Phi(\phi_k) - \nabla_\phi \hat{\Phi}_\sigma(\phi_k))||$, which is the error between the gradient of the upper objective $\nabla_\phi \Phi(\phi_k)$ and our estimate of the gradient of the pseudo objective $\nabla_\phi \hat{\Phi}_\sigma(\phi_k))$. This error is decomposed into the following terms,

- The difference between the gradient of the upper objective function $\nabla_\phi \Phi(\phi)$ and the gradient of the pseudo objective $\nabla_\phi \Phi_\sigma(\phi)$. In Kwon et al. (2024) this difference is shown to be upper bounded by $\tilde{\mathcal{O}}(\sigma)$.
- The error incurred in the estimation of the gradient terms $\nabla G(\phi, \lambda^*(\phi))$, $\nabla J(\lambda^*(\phi), \phi)$ and $\nabla J(\lambda_\sigma^*(\phi), \phi)$ given in Equation (11). The error in estimating these terms can be further broken into two error terms
  - The first source of error is in the estimation of the terms $\lambda^*(\phi)$ for the first two terms and $\lambda_\sigma^*(\phi)$ for the final term. A key insight is that both these parameters are maximizers of objective functions which follow the weak gradient domination property and this fact along with Assumption 3 id used to upper bound this error.

- The second source of error is the fact that we are taking stochastic estimates of expectations. These are bounded using Assumption 1(d) as well as Assumption 2

The details of this are given in Section B of the Appendix.

## 8 CONCLUSION

Aligning large language models (LLMs) with human values through Reinforcement Learning from Human Feedback (RLHF) is essential but faces lots of challenges, particularly due to distribution shift issues between the reward learning and policy learning stages. Existing methods often lack the theoretical foundation necessary for practical neural network-parameterized settings, limiting their effectiveness. Our work bridges this critical gap by employing a bi-level optimization framework based on the formulation in Kwon et al. (2024) and introducing the assumption of Weak Gradient Domination. This approach allows us to analyze and demonstrate convergence in an RLHF setup, achieving a sample complexity of $\epsilon^{-7/2}$. We propose a first-order method to solve the bi-level problem and provide theoretical convergence rates with state-of-the-art bounds.

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

## A    PROOF OF THEOREM 1

*Proof.*  Assuming smoothness on $\Phi$ implies the following,

$$-\Phi(\phi_{t+1}) \quad \leq \quad -\Phi(\phi_t) - \langle \nabla_\phi \Phi(\phi_t), \phi_{t+1} - \phi_t \rangle + \alpha_1 ||\phi_{t+1} - \phi_t||^2, \tag{27}$$

$$\leq \quad -\Phi(\phi_t) - \eta_t \frac{\langle \nabla_\phi \Phi(\phi_t), \nabla_\phi \hat{\Phi}_\sigma(\phi_t) \rangle}{||\nabla_\phi \hat{\Phi}_\sigma(\phi_t)||} + \alpha_1 ||\phi_{t+1} - \phi_t||^2. \tag{28}$$

Here, $\eta_t$ is the actor step size at iteration $t$ of Algorithm 1. We then define the term $e_t = \nabla_\phi \Phi(\phi_t) - \nabla_\phi \hat{\Phi}_\sigma(\phi_t)$. Here $\nabla_\phi \Phi(\phi_t)$ is the true gradient of $\Phi$ and $\nabla_\phi \hat{\Phi}_\sigma(\phi_t)$ is our estimate of $\nabla_\phi \Phi_\sigma(\phi_t)$ Consider two cases, first if $||e_t|| < \frac{1}{2} ||\nabla_\phi \Phi(\phi_t)||$, then we have

$$-\frac{\langle \nabla_\phi \Phi(\phi_t), \nabla_\phi \hat{\Phi}_\sigma(\phi_t) \rangle}{||\nabla_\phi \hat{\Phi}_\sigma(\phi_t)||} \quad = \quad \frac{-||\nabla_\phi \Phi(\phi_t)||^2 - \langle \nabla_\phi \Phi(\phi_t), e_t \rangle}{||\nabla_\phi \hat{\Phi}_\sigma(\phi_t)||}, \tag{29}$$

$$\leq \quad \frac{-||\nabla_\phi \Phi(\phi_t)||^2 + ||\nabla_\phi \Phi(\phi_t)|| \cdot ||e_t||}{||\nabla_\phi \hat{\Phi}_\sigma(\phi_t)||}, \tag{30}$$

$$\leq \quad \frac{-||\nabla_\phi \Phi(\phi_t)||^2 + ||\nabla_\phi \Phi(\phi_t)|| \cdot ||e_t||}{||\nabla_\phi \hat{\Phi}_\sigma(\phi_t)||}, \tag{31}$$

$$\leq \quad \frac{-||\nabla_\phi \Phi(\phi_t)||^2 + \frac{1}{2} ||\nabla_\phi \Phi(\phi_t)||^2}{||\nabla_\phi \hat{\Phi}_\sigma(\phi_t)||}, \tag{32}$$

$$\leq \quad -\frac{||\nabla_\phi \Phi(\phi_t)||^2}{2(||e_t|| + ||\nabla_\phi \Phi(\phi_t)||)}, \tag{33}$$

$$\leq \quad -\frac{1}{3} ||\nabla_\phi \Phi(\phi_t)||. \tag{34}$$

If $||e_t|| \geq \frac{1}{2} ||\nabla_\phi \Phi(\phi_t)||$, then we have

$$\frac{\langle \nabla_\phi \Phi(\phi_t), \nabla_\phi \hat{\Phi}_\sigma(\phi_t) \rangle}{||\nabla_\phi \hat{\Phi}_\sigma(\phi_t)||} \quad \leq \quad ||\nabla_\phi \Phi(\phi_t)||, \tag{35}$$

$$= \quad -\frac{1}{3} ||\nabla_\phi \Phi(\phi_t)|| + \frac{4}{3} ||\nabla_\phi \Phi(\phi_t)||, \tag{36}$$

$$\leq \quad -\frac{1}{3} ||\nabla_\phi \Phi(\phi_t)|| + \frac{8}{3} ||e_t||. \tag{37}$$

This technique was used in Fatkhullin et al. (2023). Now, using Equation (37) in Equation (28), we get

$$-\Phi(\phi_{t+1}) \quad \leq \quad -\Phi(\phi_t) - \frac{\eta_t}{3} ||\nabla_\phi \Phi(\phi_t)|| + \frac{8\eta_t}{3} ||e_t|| + \alpha_1 ||\phi_{t+1} - \phi_t||^2. \tag{38}$$

From the gradient domination property of $\Phi$, we have

$$-\Phi(\phi_{t+1}) \quad \leq \quad -\Phi(\phi_t) - \frac{\eta_t \sqrt{\mu_1}}{3} (\Phi^* - \Phi(\phi_t)) + \frac{8\eta_t}{3} ||\nabla_\phi \Phi(\phi_t) - \nabla_\phi \hat{\Phi}_\sigma(\phi_t)||$$
$$+ \quad \alpha_1 ||\phi_{t+1} - \phi_t||^2, \tag{39}$$

$$\Phi^* - \Phi(\phi_{t+1}) \quad \leq \quad \Phi^* - \Phi(\phi_t) - \frac{\eta_t \sqrt{\mu_1}}{3} (\Phi^* - \Phi(\phi_t)) + \frac{8\eta_t}{3} ||\nabla_\phi \Phi(\phi_t) - \nabla_\phi \hat{\Phi}_\sigma(\phi_t)||$$
$$+ \quad \alpha_1 ||\phi_{t+1} - \phi_t||^2, \tag{40}$$

$$\delta_{t+1} \quad \leq \quad \left(1 - \frac{\eta_t \sqrt{\mu_1}}{3}\right) \delta_t + \frac{8\eta_t}{3} ||\nabla_\phi \Phi(\phi_t) - \nabla_\phi \hat{\Phi}_\sigma(\phi_t)||$$
$$+ \quad \alpha_1 ||\phi_{t+1} - \phi_t||^2, \tag{41}$$

$$\leq \quad \left(1 - \frac{\eta_t \sqrt{\mu_1}}{3}\right) \delta_t + \frac{8\eta_t}{3} ||\nabla_\phi \Phi(\phi_t) - \nabla_\phi \hat{\Phi}_\sigma(\phi_t)||$$
$$+ \quad \alpha_1 \eta_t^2, \tag{42}$$

where $\delta_t = \Phi^* - \Phi(\phi_t)$. In Equation (42), if we plug in the value of $\delta_t$ and evaluate the resulting Equation for $t - 1$, we get the following.

$$\delta_{t+1} \leq \left(1 - \frac{\eta_t \cdot \sqrt{\mu_1}}{3}\right)\left(1 - \frac{\eta_t \cdot \sqrt{\mu_1}}{3}\right)\delta_{\lambda_{t-1}}$$

$$+ \left(1 - \frac{\eta_t \sqrt{\mu_1}}{3}\right)\eta_t(||\Phi(\phi_{t-1}) - d_{t-1}||+) + \eta_t(||\nabla_\phi \Phi(\phi_t) - \nabla_\phi \hat{\Phi}_\sigma(\phi_t)||+)$$

$$+ \left(1 - \frac{\eta_t \sqrt{\mu_1}}{3}\right)\alpha_1 \eta_{t-1}^2 + \alpha_1 \eta_t^2, \tag{43}$$

$$\delta_t \leq \underbrace{\Pi_{k=2}^{k=t}\left(1 - \frac{\eta_k \sqrt{\mu_1}}{3}\right)\delta_{\lambda_2}}_{A}$$

$$+ \underbrace{\sum_{k=0}^{k=t-2}\left(\Pi_{i=0}^{k-1}\left(1 - \frac{\eta_{(t-i)}\sqrt{\mu_1}}{3}\right)\right)^{\mathbb{1}(k\geq 1)}\eta_{t-k}(||\nabla_\phi \Phi(\phi_{t-k}) - \nabla_\phi \hat{\Phi}_\sigma(\phi_{t-k})||+)}_{B}$$

$$+ \underbrace{\alpha_1 \sum_{k=0}^{k=t-2}\left(\Pi_{i=0}^{i=k-1}\left(1 - \frac{\eta_{(t-i)}\sqrt{\mu_1}}{3}\right)\right)^{\mathbb{1}(k\geq 1)}(\eta_{t-k})^2}_{C}. \tag{44}$$

Let us consider the term $A$ is equation (44), if $\eta_k = \frac{\eta_1}{k}$ where $\eta_1 = \frac{7}{2\sqrt{\mu_1}}$, then we have

$$1 - \frac{\eta_k \sqrt{\mu_1}}{3} = 1 - \frac{7}{6k} \tag{45}$$

$$\leq 1 - \frac{1}{k}, \tag{46}$$

$$\leq \frac{k-1}{k}, \tag{47}$$

$$\leq \frac{k}{k-1}. \tag{48}$$

Thus, we have

$$A = \Pi_{k=2}^{k=t}\left(1 - \frac{\eta_k \sqrt{\mu_1}}{3}\right)\delta_{\lambda_2} \leq \Pi_{k=2}^{k=t}\left(\frac{\eta_k}{\eta_{k-1}}\right)\delta_{\lambda_2}, \tag{49}$$

$$\leq \frac{\eta_t}{\eta_1}\delta_{\lambda_2} = \frac{1}{t}\delta_{\lambda_2}. \tag{50}$$

Consider the term $B$ is Equation (44)

$$B = \sum_{k=0}^{k=t-2}\left(\Pi_{i=0}^{k-1}\left(1 - \frac{\eta_{(t-i)}\sqrt{\mu_1}}{3}\right)\right)^{\mathbb{1}(k\geq 1)}\eta_{t-k}(||\nabla_\phi \Phi(\phi_{t-k}) - \nabla_\phi \hat{\Phi}_\sigma(\phi_{t-k})||). \tag{51}$$

If we now consider the coefficients of $(||\nabla_\phi \Phi(\phi_{t-k}) - \nabla_\phi \hat{\Phi}_\sigma(\phi_{t-k})||)$, we see the following: For $k = 0$, the product term is 1 due to the indicator function $\mathbb{1}(k \geq 1)$.

For $k = 1$, suppose the coefficient is $\eta_k = \frac{\eta_1}{k}$. Then we have

$$\left(1 - \frac{\eta_1 \sqrt{\mu_1}}{3t}\right)\frac{\eta_1}{t-1} = \left(\frac{t - \frac{\eta_1 \sqrt{\mu_1}}{3}}{t-1}\right)\frac{\eta_1}{t}. \tag{52}$$

For $k = 2$ we have

$$\left(1 - \frac{\frac{\eta_1 \sqrt{\mu_1}}{3}}{t}\right)\left(1 - \frac{\frac{\eta_1 \sqrt{\mu_1}}{3}}{t-1}\right)\frac{\eta_1}{t-2} = \left(\frac{t - \frac{\eta_1 \sqrt{\mu_1}}{3} - 1}{t-2}\right)\left(\frac{t - \frac{\eta_1 \sqrt{\mu_1}}{3}}{t-1}\right)\frac{\eta_1}{t}. \tag{53}$$

In general, for a general $k$ this coefficient is thus

$$\Pi_{i=1}^{k} \left( \frac{t - (\frac{\eta_1 \sqrt{\mu_1}}{3} + i - 1)}{t - i} \right) \frac{\eta_1}{t}. \tag{54}$$

For $\eta_1 = \frac{7}{2\sqrt{\mu_1}}$, the numerator in all product terms is less than the denominator, hence the product term is less than 1. Therefore, all the coefficients in $B$ are upper bounded by $\eta_t$. Thus, we have

$$B \leq \frac{\eta_1}{t} \sum_{k=1}^{k=t-2} (||\nabla_\phi \Phi(\phi_k) - \nabla_\phi \hat{\Phi}(\phi_k)||), \tag{55}$$

For the term $C$ is equation (44), we have

$$C = \eta_1 \sum_{k=0}^{k=t-2} \left( \Pi_{i=0}^{i=k-1} \left( 1 - \frac{\eta_{(t-i)}\sqrt{\mu_1}}{3} \right) \right)^{\mathbb{1}(k \geq 1)} (\eta_{t-k})^2, \tag{56}$$

Similar to what was done for $A$ consider the coefficients of $\alpha_{t-k}{}^2$. For $k = 0$, the product term is 1 due to the indicator function $\mathbb{1}(k \geq 1)$. for $k = 1$ if we have $\eta_1 = \frac{7}{2\sqrt{\mu_1}}$ then

$$\left( 1 - \frac{\eta_1 \sqrt{\mu_1}}{3t} \right) \left( \frac{\eta_1}{t-1} \right)^2 \leq \left( \frac{\eta_1}{t-1} \right)^2, \tag{57}$$

for $k = 2$ if we have $\eta_1 = \frac{7}{2\sqrt{\mu_1}}$ then

$$\left( 1 - \frac{\eta_1 \mu_1}{t} \right) \left( 1 - \frac{\eta_1 \mu_1}{t-1} \right) \left( \frac{\eta_1}{t-2} \right)^2 = \frac{\left( t - \frac{\eta_1 \sqrt{\mu_1}}{3} \right)}{t} \frac{\left( t - \frac{\eta_1 \sqrt{\mu_1}}{3} - 1 \right)}{t-1} \left( \frac{\eta_1}{t-2} \right)^2, \tag{58}$$

$$\leq \left( \frac{\eta_1}{t-2} \right)^2. \tag{59}$$

This is because both terms in the coefficient of $\left( \frac{\eta_1}{t-2} \right)^2$ are less than 1. In general, for any $k$, if $\eta_1 = \frac{7}{2\sqrt{\mu_1}}$, then we have

$$\Pi_{i=0}^{i=k-1} \left( 1 - \frac{\eta_1 \mu_1}{t-i} \right) \left( \frac{\eta_1}{t-k} \right)^2 = \frac{\left( t - \frac{\eta_1 \sqrt{\mu_1}}{3} \right)}{t} \frac{\left( t - \frac{\eta_1 \sqrt{\mu_1}}{3} - 1 \right)}{t-1} \cdots$$

$$\cdots \frac{\left( t - \frac{\eta_1 \sqrt{\mu_1}}{3} - k + 1 \right)}{t-k+1} \left( \frac{\eta_1}{t-k} \right)^2,$$

$$\leq \left( \frac{\eta_1}{t-k} \right)^2. \tag{60}$$

Therefore, we have

$$C \leq \eta_1 \sum_{k=2}^{k=t-2} \left( \frac{\eta_1}{t-k} \right)^2, \tag{61}$$

$$\leq \frac{\eta_1 \cdot \eta_1^2}{t}. \tag{62}$$

We get Equation (62) from (61) by using the fact that $\sum_{k=1}^{t} \frac{1}{k^2} \leq \frac{1}{t}$. Plugging equation (50), (55), and (62) into equation (44) we get

$$\delta_t \leq \left( \frac{1}{t} \right) \delta_{\lambda_2} + \frac{\eta_1}{t} \sum_{k=0}^{k=t-2} \underbrace{(||\nabla_\phi \Phi(\phi_k) - \nabla_\phi \hat{\Phi}_\sigma(\phi_k))||)}_{A_k} + \frac{\eta_1^3}{t}, \tag{63}$$

$$\leq \frac{\eta_1}{t} \sum_{k=0}^{k=t-2} \underbrace{(||\nabla_\phi \Phi(\phi_k) - \nabla_\phi \hat{\Phi}_\sigma(\phi_k))||)}_{A_k} + \tilde{\mathcal{O}} \left( \frac{1}{t} \right). \tag{64}$$

We now bound $A_k$ as follows

$$
\begin{aligned}
||\nabla_\phi \Phi(\phi_k) - \nabla_\phi \hat{\Phi}_\sigma(\phi_k))|| &= ||\nabla_\phi \Phi(\phi_k) - \nabla_\phi \Phi_\sigma(\phi_k) + \nabla_\phi \Phi_\sigma(\phi_k) - \nabla_\phi \hat{\Phi}_\sigma(\phi_k))||, \\
&\qquad\qquad (65) \\
&\leq ||\nabla_\phi \Phi(\phi_k) - \nabla_\phi \Phi_\sigma(\phi_k))|| \\
&+ ||\nabla_\phi \Phi_\sigma(\phi_k) - \nabla_\phi \hat{\Phi}_\sigma(\phi_k))||, \qquad (66) \\
&\leq \mathcal{O}(\sigma) + \underbrace{||\nabla_\phi \Phi_\sigma(\phi_k) - \nabla_\phi \hat{\Phi}_\sigma(\phi_k))||}_{A_k'}, \qquad (67)
\end{aligned}
$$

The first term on the right hand side denotes the gap between the gradient of the objective function and the gradient of the pseudo-objective $\Phi_\sigma$. We get the upper bound on this term form Chen et al. (2024). The term $A_k'$ denotes the error incurred in estimating the true gradient of the pseudo-objective.

$$
\begin{aligned}
\underbrace{||\nabla_\phi \Phi_\sigma(\phi_k) - \nabla_\phi \hat{\Phi}_\sigma(\phi_k))||}_{A_k'} &\leq \left\| \nabla_\phi G(\phi, \lambda^*(\phi)) + \frac{\nabla_\phi J(\lambda^*(\phi), \phi) - \nabla_\phi J(\lambda_\sigma^*(\phi), \phi)}{\sigma} \right. \\
&\quad \left. - \nabla_\phi \hat{G}(\phi, \lambda_t^K) + \frac{\nabla_\phi \hat{J}(\lambda_t^K(\phi), \phi) - \nabla_\phi \hat{J}(\lambda'^{K}_t(\phi), \phi)}{\sigma} \right\|, (68) \\
&\leq ||\nabla_\phi G(\phi, \lambda^*(\phi)) - \nabla_\phi \hat{G}(\phi, \lambda_t^K)|| \\
&+ \frac{1}{\sigma}||\nabla_\phi J(\lambda^*(\phi), \phi) - \nabla_\phi \hat{J}(\lambda_t^K, \phi)|| \\
&+ \frac{1}{\sigma}||\nabla_\phi J(\lambda_\sigma^*(\phi), \phi) - \nabla_\phi \hat{J}(\lambda'^{K}_t, \phi)||. \qquad (69)
\end{aligned}
$$

As stated in the main text, the error in estimation of the gradient of the pseudo objective is split into the error in estimating $\nabla_\phi G(\phi, \lambda^*(\phi))$, $\nabla_\phi J(\lambda^*(\phi), \phi)$ and $\nabla_\phi J(\lambda_\sigma^*(\phi), \phi)$ whose respective sample based estimates are denoted by $\nabla_\phi \hat{G}(\phi, \lambda_t^K)$, $\nabla_\phi \hat{J}(\lambda_t^K, \phi)$ and $\nabla_\phi \hat{J}(\lambda'^{K}_t, \phi)$ respectively. From Lemmas 2, 3, and 4 we have

$$
\begin{aligned}
\underbrace{||\nabla_\phi \Phi_\sigma(\phi_k) - \nabla_\phi \hat{\Phi}_\sigma(\phi_k))||}_{A_k'} &\leq \tilde{\mathcal{O}}\left(\frac{1}{\sqrt{B}}\right) + \tilde{\mathcal{O}}\left(\frac{1}{\sigma K}\right) + \tilde{\mathcal{O}}\left(\frac{1}{\sigma \sqrt{n}}\right) + \tilde{\mathcal{O}}(\sqrt{\epsilon_{bias}}) \\
&+ \tilde{\mathcal{O}}(\epsilon_{approx}) + \tilde{\mathcal{O}}\left(m^{-\frac{1}{12}} D^{\frac{7}{2}}\right) \qquad (70)
\end{aligned}
$$

Plugging Equation (70) into Equation (67), then plugging the result into Equation (64) and squaring both sides we get

$$
\begin{aligned}
(\Phi^* - \Phi_t)^2 &\leq \tilde{\mathcal{O}}\left(\frac{1}{t^2}\right) + \tilde{\mathcal{O}}\left(\frac{1}{\sigma^2 K^2}\right) + \tilde{\mathcal{O}}\left(\frac{1}{\sigma^2 n}\right) + \tilde{\mathcal{O}}(\epsilon_{bias}) + \tilde{\mathcal{O}}(\epsilon_{approx}^2) \\
&+ \tilde{\mathcal{O}}\left(\frac{1}{B}\right) + \tilde{\mathcal{O}}(\sigma^2) + \tilde{\mathcal{O}}\left(m^{-\frac{1}{6}} D^7\right) \qquad (71)
\end{aligned}
$$

$\square$

## B   SUPPLEMENTARY LEMMAS

**Lemma 2.** *For a fixed iteration $K$ of Algorithm 1 and $\phi \in \Theta$ we have*

$$
\begin{aligned}
||\nabla G(\phi, \lambda^*(\phi)) - \nabla_\phi \hat{G}(\phi, \lambda_t^K)|| &\leq \tilde{\mathcal{O}}\left(\frac{1}{\sqrt{B}}\right) + \tilde{\mathcal{O}}\left(\frac{1}{K}\right) + \tilde{\mathcal{O}}\left(\frac{1}{\sqrt{n}}\right) \\
&+ \tilde{\mathcal{O}}\left(m^{-\frac{1}{12}} D^{\frac{7}{2}}\right) + \tilde{\mathcal{O}}(\sqrt{\epsilon_{bias}}) + \tilde{\mathcal{O}}(\epsilon_{approx}).
\end{aligned}
$$

*Proof.*

$$
\begin{aligned}
||\nabla_\phi G(\phi, \lambda^*(\phi)) - \nabla_\phi \hat{G}(\phi, \lambda_t^K)|| &\leq ||\nabla_\phi G(\phi, \lambda^*(\phi)) - \nabla_\phi G(\phi, \lambda_t^K) \\
&+ \nabla_\phi G(\phi, \lambda_t^K) - \nabla_\phi \hat{G}(\phi, \lambda_t^K)||, \qquad (72) \\
&\leq \underbrace{||\nabla_\phi G(\phi, \lambda^*(\phi)) - \nabla_\phi G(\phi, \lambda_t^K)||}_{A_k'} \\
&+ \underbrace{||\nabla_\phi G(\phi, \lambda_t^K) - \nabla_\phi \hat{G}(\phi, \lambda_t^K)||}_{B_k'}. \qquad (73)
\end{aligned}
$$

Here $A_k'$ represents the error incurred in due to difference between $\lambda^*(\phi)$ and our estimate $\lambda_t^K$. The term $B_k'$ represents the difference between the true gradient $\nabla_\phi G(\phi, \lambda_t^K)$ and its sample based estimate. We first bound $A_k'$.

$$
\begin{aligned}
||\nabla_\phi G(\phi, \lambda^*(\phi)) - \nabla_\phi G(\phi, \lambda_t^K)|| &\leq L||\lambda^*(\phi) - \lambda_t^K|| \qquad (74) \\
&\leq L_1 \cdot \lambda' ||J(\lambda^*(\phi)) - J(\lambda_t^K, \phi))||. \qquad (75)
\end{aligned}
$$

Here $L_1$ is the smoothness constant of $G(\lambda, \phi)$. We get Equation (75) from Equation (74) by Assumption 3. Now, consider the function $J(\lambda, \phi)$. We know from Lemma 1 that it satisfies the weak gradient condition, therefore applying the same logic for $J(\lambda, \phi)$ that we did for $\Phi(\sigma)$. Similar to the result in Equation (64), we obtain

$$
\begin{aligned}
&J(\lambda^*(\phi)) - J(\lambda_t^K, \phi)) \\
&\leq \frac{\tau_1}{K} \sum_{i=0}^{i=K-2} \underbrace{(||\nabla_\lambda J(\lambda_t^i, \phi) - d_t)||)}_{A'} + \tilde{\mathcal{O}}\left(\frac{1}{K}\right), \qquad (76) \\
&\leq \tilde{\mathcal{O}}\left(\frac{1}{K}\right) + \frac{\tau_1 . M_g}{K} \sum_{i=0}^{i=K-2} \mathbb{E}_{(s,a) \sim d_\nu^{\pi_{\lambda_i}}} |Q_\phi^{\pi_{\lambda_i}}(s,a) - Q_{i,J}(s,a)|, \\
&\qquad\qquad\qquad\qquad\qquad\qquad\qquad\qquad\qquad\qquad\qquad\qquad\qquad (77) \\
&\leq \tilde{\mathcal{O}}\left(\frac{1}{K}\right) + \tilde{\mathcal{O}}\left(\frac{1}{\sqrt{n}}\right) + \tilde{\mathcal{O}}\left(m^{-\frac{1}{12}} D^{\frac{7}{2}}\right) + \tilde{\mathcal{O}}(\epsilon_{approx}) \\
&+ \tilde{\mathcal{O}}(\sqrt{\epsilon_{bias}}). \qquad (78)
\end{aligned}
$$

We get Equation (77) from Equation (76) by using Equations 81-86 from Gaur et al. (2024) and we get Equation (78) from Equation (77) from Equation 99 of Gaur et al. (2024). We now bound $B_k'$ as

$$
\begin{aligned}
&||\nabla_\phi G(\phi, \lambda_t^K) - \nabla_\phi \hat{G}(\phi, \lambda_t^K)|| \\
&= \mathbb{E}\sqrt{d \cdot \sum_{p=1}^{d}\left(\left(\sum_{i=1}^{B}\frac{1}{B}\nabla_\phi \hat{G}_{\tau_i}(\phi, \lambda_t^K)\right)_p - \left(\sum_{i=1}^{B}\frac{1}{B}\mathbb{E}_\tau \nabla_\phi \hat{G}_{\tau_i}(\phi, \lambda_t^K)\right)_p\right)^2}, \\
&\qquad\qquad\qquad\qquad\qquad\qquad\qquad\qquad\qquad\qquad\qquad\qquad\qquad (79) \\
&= \sqrt{\frac{d}{B^2}\cdot\sum_{p=1}^{d}\mathbb{E}\left(\sum_{i=1}^{B}\left(\nabla_\phi \hat{G}_{\tau_i}(\phi, \lambda_t^K)_p - \mathbb{E}_\tau \nabla_\phi \hat{G}_{(\tau_i)}(\phi, \lambda_t^K)_p\right)\right)^2}, \qquad (80) \\
&\leq \sqrt{\frac{d^2 . B . \sigma_G}{B^2}}, \qquad (81) \\
&\leq \sqrt{d . \frac{\sigma_G}{B}}. \qquad (82)
\end{aligned}
$$

Here, the right hand side of Equation (79) comes from writing out the definition of the $l_1$ norm where the subscript of $p$ denotes the $p^{th}$ co-ordinate of the gradient. In Equation (80), we take the

expectation with respect to the pair of trajectories and their corresponding preferences on both sides of the equation. Equation (81) is obtained from Equation (80) by using Jensen's Inequality and Equation (82) is obtained from 81 using Assumption 1 which bounds the variance of the estimate of $\nabla_\phi G(\phi, \lambda_t^K)$. Plugging Equation (78) and 82 into Equation (73), we get the required result. $\qquad\square$

**Lemma 3.** *For a fixed iteration $K$ of Algorithm 1 and $\phi \in \Theta$, we have*

$$
\begin{aligned}
||\nabla_\phi J(\lambda_\sigma^*(\phi), \phi) - \nabla_\phi \hat{J}(\lambda_t^K(\phi), \phi)|| &\leq \tilde{\mathcal{O}}\left(\frac{1}{B}\right) + \tilde{\mathcal{O}}\left(\frac{1}{K}\right) + \tilde{\mathcal{O}}\left(\frac{1}{\sqrt{n}}\right) \\
&+ \tilde{\mathcal{O}}(\sqrt{\epsilon_{bias}}) + \tilde{\mathcal{O}}(\epsilon_{approx})
\end{aligned} \tag{83}
$$

*Proof.*

$$
||\nabla_\phi J(\lambda(\phi), \phi) - \nabla_\phi \hat{J}(\lambda_t^K(\phi), \phi)||
$$

$$
\leq ||\nabla_\phi J(\lambda(\phi), \phi) - \nabla_\phi J(\lambda_t^K, \phi) + \nabla_\phi J(\lambda_t^K, \phi)\nabla_\phi - \hat{J}(\lambda_t^K(\phi), \phi)||, \tag{84}
$$

$$
\leq ||\nabla_\phi J(\lambda(\phi), \phi) - \nabla_\phi J(\lambda_t^K, \phi)|| + ||\nabla_\phi J(\lambda_t^K, \phi)\nabla_\phi \hat{J}(\lambda_t^K(\phi), \phi)||, \tag{85}
$$

$$
\leq ||(\lambda^*(\phi)) - (\lambda_t^K)|| + ||\nabla_\phi J(\lambda_t^K, \phi) - \nabla_\phi \hat{J}(\lambda_t^K(\phi), \phi)||, \tag{86}
$$

$$
\leq \underbrace{L'||J(\lambda^*, \phi) - J(\lambda_t^K, \phi)||}_{A''} + \underbrace{||\nabla_\phi J(\lambda_t^K, \phi) - \nabla_\phi \hat{J}(\lambda_t^K(\phi), \phi)||}_{B''}. \tag{87}
$$

We get Equation (86) form Equation (85) by using Assumption 3. The first term $A''$ is upper bounded in the exact same manner as in Lemma 2. Thus, we have

$$
\begin{aligned}
J(\lambda^*(\phi), \phi) - J(\lambda_t^K, \phi)) &\leq \tilde{\mathcal{O}}\left(\frac{1}{K}\right) + \tilde{\mathcal{O}}\left(\frac{1}{\sqrt{n}}\right) + \tilde{\mathcal{O}}\left(m^{-\frac{1}{12}}D^{\frac{7}{2}}\right) + \tilde{\mathcal{O}}(\sqrt{\epsilon_{bias}}) \\
&+ \tilde{\mathcal{O}}(\epsilon_{approx}).
\end{aligned} \tag{88}
$$

We bound $B''$ as follows

$$
||\nabla_\phi J(\lambda_t^K, \phi) - \nabla_\phi \hat{J}(\lambda_t^K(\phi), \phi)||
$$

$$
= \left|\left|\frac{1}{B}\sum_{j=1}^{B}\sum_{i=1}^{\infty}\gamma^{i-1}\mathbb{E}[\nabla_\phi r_\phi(s_{i,j}, a_{i,j})] - \frac{1}{B}\sum_{j=1}^{B}\sum_{i=1}^{H}\gamma^{i-1}\nabla_\phi r_\phi(s_{i,j}, a_{i,j})\right|\right|
$$

$$
\leq \left|\left|\sum_{j=1}^{B}\left(\sum_{i=1}^{\infty}\gamma^{i-1}\mathbb{E}[\nabla_\phi r_\phi(s_{i,j}, a_{i,j})] - \sum_{i=H+1}^{\infty}\gamma^{i-1}\mathbb{E}\nabla_\phi r_\phi(s_{i,j}, a_{i,j}),\right.\right.\right. \tag{89}
$$

$$
\left.\left.\left.+ \sum_{i=H+1}^{\infty}\gamma^{i-1}\mathbb{E}[\nabla_\phi r_\phi(s_{i,j}, a_{i,j})] - \sum_{i=1}^{H}\gamma^{i-1}\nabla_\phi r_\phi(s_{i,j}, a_{i,j})\right)\right|\right|, \tag{90}
$$

$$
\leq \left|\left|\frac{1}{B}\left(\sum_{i=1}^{H}\gamma^{i-1}(\mathbb{E}[\nabla_\phi r_\phi(s_{i,j}, a_{i,j})] - \nabla_\phi r_\phi(s_{i,j}, a_{i,j}))\right)\right|\right|
$$

$$
+ \left|\left|\frac{1}{B}\sum_{j=1}^{B}\sum_{i=H}^{\infty}\gamma^{i-1}\mathbb{E}[\nabla_\phi r_\phi(s_{i,j}, a_{i,j})]\right|\right|, \tag{91}
$$

$$
\leq \tilde{\mathcal{O}}\left(\frac{\gamma^H}{B}\right) + \tilde{\mathcal{O}}(\gamma^H), \tag{92}
$$

The terms on the right hand side of Equation (91) is upper bounded from Assumption 2. Plugging Equation (92) and (88) into Equation (87) gives us the required result. $\qquad\square$

**Lemma 4.** *For a fixed iteration $K$ of Algorithm 1 we have the following*

$$||\nabla_\phi J(\lambda_\sigma^*(\phi), \phi) - \nabla_\phi \hat{J}(\lambda_t^{'K}(\phi), \phi)|| \leq \tilde{\mathcal{O}}\left(\frac{1}{\sqrt{B}}\right) + \tilde{\mathcal{O}}\left(\frac{1}{K}\right) + \tilde{\mathcal{O}}\left(\frac{1}{\sqrt{n}}\right)$$
$$+ \tilde{\mathcal{O}}(\sqrt{\epsilon_{bias}}) + \tilde{\mathcal{O}}(\epsilon_{approx}) \tag{93}$$

*Proof.*

$$||\nabla_\phi J(\lambda_\sigma^*(\phi), \phi) - \nabla_\phi \hat{J}(\lambda_t^{'K}(\phi), \phi)||$$

$$\leq ||\nabla_\phi J(\lambda_\sigma^*(\phi), \phi) - \nabla_\phi J(\lambda_t^{'K}, \phi) + \nabla_\phi J(\lambda_t^{'K}, \phi) \nabla_\phi \hat{J}(\lambda_t^{'K}(\phi), \phi)||, \tag{94}$$

$$\leq ||\nabla_\phi J(\lambda_\sigma^*(\phi), \phi) - \nabla_\phi J(\lambda_t^{'K}, \phi)|| + ||\nabla_\phi J(\lambda_t^{'K}, \phi) \nabla_\phi \hat{J}(\lambda_t^{'K}(\phi), \phi)||, \tag{95}$$

$$\leq ||(\lambda_\sigma^*(\phi)) - (\lambda_t^{'K})|| + ||\nabla_\phi J(\lambda_t^{'K}, \phi) - \nabla_\phi \hat{J}(\lambda_t^{'K}(\phi), \phi)||, \tag{96}$$

$$\leq \underbrace{L'||J_\sigma(\lambda_\sigma^*, \phi) - J_\sigma(\lambda_t^{'K}, \phi)||}_{A''} + \underbrace{||\nabla_\phi J(\lambda_t^{'K}, \phi) - \nabla_\phi \hat{J}(\lambda_t^{'K}(\phi), \phi)||}_{B''}. \tag{97}$$

We get Equation (97) from Equation (96) using Assumption 3. Note that $B''$ here is the same as in Lemma 3. Thus we have

$$||\nabla_\phi J(\lambda_t^{'K}, \phi) - \nabla_\phi \hat{J}(\lambda_t^{'K}(\phi), \phi)|| \leq \tilde{\mathcal{O}}\left(\frac{\gamma^H}{B}\right) + \tilde{\mathcal{O}}(\gamma^H) \tag{98}$$

For $A''$ note that now the gradient descent is happening on the obejctive given by $J_\sigma = J(\lambda, \phi) + \sigma G(\phi, \lambda)$. Applying the same logic as we did for $J(\lambda, \phi)$, we get

$$J_\sigma(\lambda_\sigma^*, \phi) - J_\sigma(\lambda_t^{'K}, \phi) \leq \tilde{\mathcal{O}}\left(\frac{1}{K}\right) + \frac{\tau'}{k} \sum_{i=0}^{i=k-2} \underbrace{(||\nabla_\lambda J_\sigma(\lambda_t^{'i}, \phi) - d_i'||)}_{A'}. \tag{99}$$

Now, consider the term $A'$

$$||\nabla_\lambda J_\sigma(\lambda_t^{'k}, \phi) - d_t'|| = ||\nabla_\lambda J(\lambda_t^{'k}, \phi) + \sigma \nabla_\lambda G(\lambda_t^{'k}, \phi) - d_t'||, \tag{100}$$

$$\leq \underbrace{||\nabla_\lambda J(\lambda_t^{'k}, \phi) - \frac{1}{n}\sum_{i=1}^{n}\nabla\log(\pi(a_i|s_i))Q_{k,J}(s_i, a_i)||}_{A'''}$$

$$+ \sigma \underbrace{||\nabla_\lambda G(\lambda_t^{'k}, \phi) - \frac{1}{B}\sum_{i=1}^{B}\nabla_\lambda \hat{G}_{\tau_i}(\lambda_t^{'k}, \phi_t)||}_{B'''}. \tag{101}$$

The term $A'''$ is identical to the term $A'$ in Equation (76). We then evaluate $B'''$.

$$||\nabla_\lambda G(\phi, \lambda_t^K) - \nabla_\lambda \hat{G}(\phi, \lambda_t^K)||$$

$$= \sqrt{d \cdot \sum_{p=1}^{d}\left(\left(\sum_{i=1}^{B}\frac{1}{B}\nabla_\lambda \hat{G}_{\tau_i}(\phi, \lambda_t^K)\right)_p - \mathbb{E}\left(\sum_{i=1}^{B}\frac{1}{B}\nabla_\lambda \hat{G}_{\tau_i}(\phi, \lambda_t^K)\right)_p\right)^2}$$
$$\tag{102}$$

$$= \sqrt{\frac{d}{B^2} \cdot \sum_{p=1}^{d}\left(\sum_{i=1}^{B}(U(\phi) \cdot F(\lambda))_p - \mathbb{E}[(U(\phi) \cdot F(\lambda))_p]\right)^2}, \tag{103}$$

$$\leq \sqrt{\frac{d^2 \cdot B \cdot M_G}{B^2}}, \tag{104}$$

$$\leq \sqrt{d \cdot \frac{M_G}{B}}. \tag{105}$$

Here, $F(\lambda) = \sum_{h=0}^{H_u-1} \nabla_\lambda \log\pi_\lambda(a_h|s_h)$ and the subscript $p$ denotes the $p^{th}$ co-ordinate of the gradient. We get Equation (104) from 103 from Assumption (2) and the fact that $U(\phi)$ is less than 1 for all $\phi$. Thus right hand side of Equation (97) is the same as the right hand side of Equation in Lemma (3) but with $\mathcal{O}\left(\frac{1}{\sqrt{B}}\right)$ instead of $\mathcal{O}\left(\frac{1}{B}\right)$, which is the required result. $\qquad\square$

**Lemma 5.** *For a given $\lambda \in \Lambda$ and $\phi \in \Theta$ we have*

$$\nabla_\phi J_\phi^\lambda = \sum_{i=1}^\infty \gamma^{i-1} \mathbb{E} \nabla_\phi r_\phi(s_i, a_i) \tag{106}$$

*Proof.* We start by writing the gradient of $J(\lambda, \phi)$ with respect to $\phi$ as follows

$$\nabla_\phi J(\lambda, \phi)$$

$$= \nabla_\phi \int_{s_1, a_1} Q_\phi^\lambda(s_1, a_1) \pi_\lambda(a_1|s_1) d(s_1) \tag{107}$$

$$= \int_{s_1, a_1} \nabla_\phi r_\phi(s_1, a_1) \pi_\lambda(a_1|s_1) d(s_1)$$

$$+ \gamma \cdot \nabla_\phi \int_{s_1, a_1} \int_{s_2, a_2} Q_\phi^\lambda(s_2, a_2) d(s_2|a_1) \pi_\lambda(a_2|s_2) d(s_1) \pi_\lambda(a_1|s_1), \tag{108}$$

$$= \int_{s_1, a_1} \nabla_\phi r_\phi(s_1, a_1) \pi_\lambda(a_1|s_1) d(s_1)$$

$$+ \gamma \cdot \int_{s_2, a_2} \int_{s_1, a_1} \nabla_\phi r_\phi(s_2, a_2) d(s_2|a_1) \pi_\lambda(a_2|s_2) d(s_1) \pi_\lambda(a_1|s_1)$$

$$+ \gamma^2 \cdot \nabla_\phi \int_{s_1, a_1} \int_{s_2, a_2} \int_{s_3, a_3} Q_\phi^\lambda(s_3, a_3) d(a_3|s_3) d(s_3|a_2) d(s_2|a_1) \pi_\lambda(a_2|s_2) d(s_1) \pi_\lambda(a_1|s_1), \tag{109}$$

$$= \int_{s_1, a_1} \nabla_\phi r_\phi(s_1, a_1) d(s_1, a_1)$$

$$+ \gamma \cdot \int_{s_2, a_2} \nabla_\phi r_\phi(s_2, a_2) d(s_2, a_3) + \gamma^2 \cdot \nabla_\phi \int_{s_3, a_3} Q_\phi^\lambda(s_3, a_3) d(s_3, a_3). \tag{110}$$

We get Equation (108) from Equation (107) by noting that $Q_\phi^\lambda(s, a) = r_\phi + \int_{s', a'} Q_\phi^\lambda(s', a') d(s'|a) \pi_\lambda(a'|s')$. We repeat the same process on the second term on the right hand side of Equation (108) to obtain Equation (109). Continuing this sequence, we get

$$\nabla_\phi J_\phi^\lambda = \sum_{i=1}^\infty \gamma^{i-1} \mathbb{E} \nabla_\phi r_\phi(s_i, a_i) \tag{111}$$

Here, $s_i, a_i$ belong to the distribution of the $i^{th}$ state action pair induced by following the policy $\lambda$. $\qquad\square$

