# OpenReview forum: "On the Global Convergence of RLHF Based Alignment With Neural Parametrization"
_ICLR.cc/2025/Conference — ICLR 2025 Conference Withdrawn Submission_

### Official Review · Reviewer_9Hek · 2024-11-02

**Soundness:** 3
**Presentation:** 3
**Contribution:** 2
**Rating:** 5
**Confidence:** 2

**Summary:**

This paper studies the global convergence of RLHF method for LLMs. The authors employ a bi-level optimization framework to address the hyper-gradient problem distributional shifts between reward learning and policy learning phases. They develop the theory based on the weak gradient domination. They introduce a first-order approach under neural network parameterization, moving away from the tabular assumptions in previous work. The study offers theoretical contributions by establishing convergence rate bounds and analyzing sample complexity for this parameterized RLHF, achieving a sample complexity of $\epsilon^{-7/2}$.
​

**Strengths:**

* The paper proposes a structured theoretical framework for the analysis of RLHF with neural networks, which extends previous studies on RLHF to the bi-level optimization framework with neural networks.

* The paper is well organized with a clear introduction to the theory framework and the derivation of the bi-level optimization framework. The proof sketch is easy to follow.

**Weaknesses:**

* Limited theoretical contribution:
  * The proposed bi-level formulation is largely based on prior work by Chakraborty et al. (2024c), Ding et al. (2024a), and Kwon et al. (2024). While the authors apply these frameworks to their RLHF problem, the innovation here is minimal, as the formulation directly follows existing works.
  * The main proof draws heavily from existing analysis techniques, integrating RL method analyses from Xu et al. (2020) with bi-level optimization results by Chen et al. (2024). Additionally, the use of NTK-based analysis follows Wang et al. (2020) for shallow and over-parameterized feed-forward neural networks. Thus, the theoretical contribution of the paper appears to be incremental.

* The analysis is limited to shallow and over-parameterized feed-forward neural networks. This does not align well with standard RLHF in LLMs, where large-scale and complex decoder-only transformers are used.

* The absence of empirical studies is a significant limitation. While the paper proposes a new algorithm for RLHF and claims that the method handles the distribution shift problem in RLHF, there is no empirical evidence supporting such claims.

**Questions:**

* Is there any theoretical or empirical evidence supporting the claim that the bi-level optimization framework handles the distribution shift in RLHF?

---

### Official Review · Reviewer_LD1r · 2024-11-02

**Soundness:** 1
**Presentation:** 2
**Contribution:** 2
**Rating:** 3
**Confidence:** 4

**Summary:**

The paper provides a new convergence analysis for bilevel RLHF framework with $\mathcal{O}(\epsilon^{-7/2})$ rate of convergence.

**Strengths:**

The theory of the paper is well-rounded and novel since it is one of the first to consider the penalty based method for solving the bilevel RLHF problem.

**Weaknesses:**

(Please answer to the "Questions" section directly) The formulation and contributions of the work seem limited. The proposed framework is lack of real applicability.

**Questions:**

I have the following concerns and questions toward this work:

1. First, I think the bilevel framework proposed and studied in [1,2,3] (and more works) actually contains a KL regularized loss in the lower-level, such as (4) in [2]. Specifically, if we add a KL-regularization to the lower-level objective, i.e. equation (6) in the paper, then the lower-level enjoys a closed form solution and may not need to be written as bilevel problem at the first place. So basically the bilevel problem that people are considering are a combination of (1) and (2) in the paper, where (2) is in the lower level and enjoys closed form solution. This greatly challenges the motivation of the work.
2. To my understanding, the penalty reformulated bilevel approaches is adapted directly from [4]. A major concern is that Algorithm 1 is not practical for LLM and RLHF pipeline. It's also not very clear if it can be applied to large-scale RL applications. On the other hand, Algorithm 2 is also a relative standard way to estimate the fitted Q-value function. In general, I think the algorithmic contribution of the work is also limited.
3. In the convergence analysis, it remains un-clear how the assumptions are applicable. Specifically, I'm wondering if the weak gradient domination assumptions in Assumption 1 are applicable for LLMs or at least two layer NNs. Otherwise the assumptions seem artificial and less useful. Same for Assumption 2 and 3.
4. Too many typos in the theoretical results. For example, Line 389 in Assumption 3, should be $J(\lambda,\phi)$ than $J(\lambda.\phi)$; Line 400 in Assumption 4, an extra period mark in front of the inequality sign; Line 426 in Lemma 1, what is $J(\lambda),\phi)$?; In Theorem 1, the exponentials are all outside of the big $O$ notation, such as $K=\tilde{\mathcal{O}}(\epsilon)^{-1}$ which are all inappropriate. There should be more in the appendix but I don't have enough time to check all the details.

In conclusion, this paper needs major revisions and improvements before publishing.

References:

[1] Chakraborty, Souradip, et al. "PARL: A unified framework for policy alignment in reinforcement learning." arXiv preprint arXiv:2308.02585 (2023): 3.

[2] Ding, Mucong, et al. "Sail: Self-improving efficient online alignment of large language models." arXiv preprint arXiv:2406.15567 (2024).

[3] Li, Chenliang, et al. "Joint Demonstration and Preference Learning Improves Policy Alignment with Human Feedback." arXiv preprint arXiv:2406.06874 (2024).

[4] Kwon, Jeongyeol, et al. "On penalty methods for nonconvex bilevel optimization and first-order stochastic approximation." arXiv preprint arXiv:2309.01753 (2023).

---

### Official Review · Reviewer_rv3w · 2024-11-04

**Soundness:** 3
**Presentation:** 1
**Contribution:** 2
**Rating:** 5
**Confidence:** 2

**Summary:**

This paper addresses challenges in Reinforcement Learning from Human Feedback (RLHF) for aligning large language models (LLMs) with human preference. The authors propose a bi-level optimization framework to address the distribution shift between reward learning and policy learning stages, which can impact the effectiveness of AI alignment. Prior work of RLHF methods generally assume a tabular or linear setting, which does not fully apply to modern neural network-parameterized LLMs.

**Strengths:**

The authors provide thorough theoretical support, including convergence rate bounds and global optimality proofs. The sample complexity achieved $\epsilon^{-7/2}$ is a notable improvement for the RLHF framework, and these state-of-the-art bounds provide a solid foundation for future empirical work and practical applications.

**Weaknesses:**

- **Clarity:**  The paper’s writing quality could be improved for clarity.  The specific form of the neural network parameterization is unclear, and the choice of activation function is not defined. In Section 4, the concept of distribution shift is not clearly explained, which is an important motivation for the proposed method. Additionally, some sentences are difficult to follow, and the extensive use of complex notation can make it easy for readers to lose track of meanings. A summary table of notations would be helpful.
- **Assumption**: The assumptions used in the actor-critic and bi-level optimization methods are all presented in the main text, making it challenging for readers to follow. Moving some of the less critical assumptions to the appendix and providing detailed explanations for the key assumptions would give readers a stronger foundation for understanding the main results.
- **Theoretical results:** The main results in Section 6.3 lack sufficient explanation, making it difficult to understand their implications. Additionally, the paper initially compares different RLHF methods' sample complexities but only discusses comparisons with actor-critic algorithms in the presentation of main results, which can be confusing for readers unfamiliar with the differences.

**Questions:**

Q1:   While theoretically sound, the bi-level optimization framework may pose challenges in practical implementation, particularly in tuning parameters for convergence. Providing practical guidelines or heuristics for selecting parameters, such as the gradient step size, would make the approach more accessible to practitioners and facilitate its application in real-world scenarios.

Q2: In Section 3, policy fine-tuning is defined as a KL-regularized policy optimization problem, but in the proposed algorithm, the KL regularization is not implemented. Could you clarify if and how KL regularization is used in the algorithm?

Q2: The current theoretical results seem heavily based on the actor-critic framework from [1], with the main adjustment being the substitution of policy optimization by solving Equation (10). Could you elaborate on the differences and challenges associated with this analysis?

Q3: The reliance on weak gradient domination and neural parameterization assumptions may limit the generalizability of the approach, as these conditions may not hold in complex, real-world environments. Could you provide further discussion on this?



[1] Closing the Gap: Achieving Global Convergence (Last Iterate) of Actor-Critic under Markovian Sampling with Neural Network Parametrization. ICML 2024.

---

### Official Review · Reviewer_jaba · 2024-11-04

**Soundness:** 2
**Presentation:** 2
**Contribution:** 2
**Rating:** 3
**Confidence:** 3

**Summary:**

This paper studies the problem of reinforcement learning with human feedback. It proposes to formulate the problem as a bi-level optimization, where the upper level optimizes a reward function, and the lower level learns the optimal policy with respect to this reward function. Under the so-called weak gradient domination assumptions, the paper proves a sample complexity of $\epsilon^{-7/2}$.

**Strengths:**

1) This paper is well-written and easy to follow in the first sections, but is poorly written in the subsequent sections (from Section 5).
2) It seems promising to provide a sample complexity guarantee for the bi-level optimization and develop an algorithm.

**Weaknesses:**

1. The theory requires strong assumptions that are hard to justify in practice. For example, Assumptions 1 and 3 regarding weak gradient domination require justification even for simple neural networks. Without this, the derived theoretical results are not meaningful.
2. The significance of the distribution shift is unclear. Current practical seems easy to implement and work well. This paper argues that the distribution shift is significant in RLHF; however, it does not provide concrete numerical evidence or theoretical justification.

**Questions:**

This paper requires a thorough revision. Currently, from Section 4 onward, it is hard to follow.

- In Section 3, it is mentioned that the RLHF problem is a special RL task with deterministic transitions and trajectory-level reward. However, later, a complicated MDP formulation is studied. The additional notations and symbols impose a significant burden on readers in the field of RLHF. It is suggested to study the simplest case where $H=1$ to illustrate the key idea of this paper.

- In the second part of Section 3, the significance of the distribution shift is not clearly defined or explained.
- In Section 4, the proxy objective requires an explanation, even if it is from prior literature. In particular, it seems that $\lambda^{*}$ is required in (10), making this proxy objective still impractical. It is unclear how the proposed formulation bypasses the difficulty of requiring the calculation of the hyper-gradient (and possibly the Hessian).
- Notations in Section 6 are poorly explained, and many do not follow proper LaTeX formatting. For example, sigma in Assumption 1, Log in Assumption 2 (i) and (ii), norm in Assumption 3, and bracket in Lemma 1.

Comments about the theory:

- The reviewer notes that Assumption 2 (i.e., Lipschitz and smoothness) may be relatively easy to satisfy. However, how can the assumptions in Assumptions 1 and 3 be met? Since this paper focuses on neural parameterization, it should discuss how neural networks, like Transformers, can meet these assumptions.
- For the main result in Theorem 1: What is the dependence on the horizon $\gamma$ and the state space size (or vocabulary size in the language model setting)?
- The sample complexity of the so-called online RLHF is not defined in this paper. In Theorem 1, the squared distance is used. It is unclear if the other results in the references in Table 1 follow this notion and whether the results listed in the Table are truly comparable.
- What would be different if the algorithm does not follow the hyper-gradient formulation?

- Could you provide numerical results to validate the superiority of the proposed algorithm and highlight the significance of the research? Currently, the theory offers limited implications, and the significance remains unclear.
- The difference from related work, *PARL: A Unified Framework for Policy Alignment in Reinforcement Learning from Human Feedback,* requires a more detailed discussion.

---

### Note · Authors · 2024-11-23

**Comment:**

We want to thank the authors for their feedback and suggestions. We will incorporate their suggestions and improve our work.

**Withdrawal Confirmation:**

I have read and agree with the venue's withdrawal policy on behalf of myself and my co-authors.